# Design and application of α-ketothioesters as 1,2-dicarbonyl-forming reagents

Ming Wang [1], Zhihong Dai[1] & Xuefeng Jiang [1,2]

The 1,2-dicarbonyl motif is vital to biomolecules, especially natural products and pharmaceuticals. Conventionally, 1,2-dicarbonyl compounds are prepared via an α-keto acyl chloride. Based on the methods used in nature, a transition-metal-free approach for the synthesis of an α-ketothioester reagent via the combination of an α-hydroxyl ketone, elemental sulfur and a benzyl halide is reported. Mechanistic studies demonstrate that the trisulfur radical anion and the α-carbon radical of the α-hydroxy ketone are involved in this transformation. The dicarbonylation of a broad range of amines and amino acids, and importantly, cross couplings with aryl borates to construct dicarbonyl-carbon bonds are realized under mild conditions by employing this stable and convenient α-ketothioester as a 1,2-dicarbonyl reagent. The dicarbonyl-containing drug indibulin and the natural product polyandrocarpamide C, which possess multiple heteroatoms and active hydrogen functional groups, can be efficiently prepared using the designed 1,2-dicarbonyl reagent.

---

[1] Shanghai Key Laboratory of Green Chemistry and Chemical Process, School of Chemistry and Molecular Engineering, East China Normal University, 3663 North Zhongshan Road, Shanghai 200062, China. [2] State Key Laboratory of Elemento-Organic Chemistry, Nankai University, Tianjin 300071, China. Correspondence and requests for materials should be addressed to X.J. (email: xfjiang@chem.ecnu.edu.cn)

The 1,2-dicarbonyl motif is an important life-related structure that is ubiquitous in natural products[1-5] and modern pharmaceuticals[6-10]. Licoagrodione, isolated from a *Chinese herb*, was found to exhibit antimicrobial activity[2]. Tanshinone IIA is a transcription factor inhibitor and was isolated from Salvia miltiorrhiza BUNGE[3]. Mansonone C, isolated from *Mansonia altissima*, displays antifungal activity against P. parasitica[4]. Sophoradione was isolated from the roots of S. flavescens and is cytotoxic to KB tumour cells (Fig. 1a)[5]. Since the dicarbonyl motif can bind to proteins in the body to increase their bioavailability, many well-known dicarbonyl-containing molecules have been turned into clinically used drugs, such as the anticancer drugs indibulin[7] and biricodar[8], the anti-HCV drug boceprevir[9], and the dermatologic agent fluocortin butyl, a synthetic corticosteroid with high topical to systemic activites (Fig. 1b)[10]. Furthermore, dicarbonyl-containing compounds frequently serve as valuable synthetic intermediates and precursors in organic synthesis and materials science. For example, aromatic substituted quinoxalines, which possess broad applications as photoinitiators and fluorescence-based sensors, have been synthesized from dicarbonyl-containing compounds[11-14]. Conventionally, dicarbonyl compounds are prepared via Müller's α-keto acyl chloride, but the compatibility is imperfect and side reactions can occur[15-18]. Based on the methods used in nature, ester bonds can be formed through the transesterification of thioesters, such as acetyl coenzyme A[19], and native chemical ligation via peptide chemistry[20]. Thioesters, as active but stable esters, have been widely used as synthetic intermediates for acyl transfer reactions such as Corey-Nicolaou macrolactonizations (Fig. 2a)[21]. Due to the C–S bond possessing both weaker bond energy and relative stability at ambient conditions, we assume that an α-ketothioester will be an excellent 1,2-dicarbonyl-forming reagent and be broadly applicable in chemistry (Fig. 2b). Previously, we found that α-hydroxy ketones were efficient acylating reagents, and that they easily initiated radical formation at the α position[22]. As a continuation of our investigations of the transformations of inorganic sulfur compounds to organic sulfur structures[23-30], we hypothesize that trisulfur radical anions can react at the α position of α-hydroxy ketones (Fig. 2c). Herein, a transition-metal-free approach for the synthesis of an α-ketothioester reagent via the combination of an α-hydroxyl ketone, elemental sulfur and a benzyl halide is reported. The dicarbonylation of a broad range of amines and amino acids, and cross couplings with aryl borates to construct dicarbonyl-carbon bonds are realized by employing this stable and convenient α-ketothioester as a 1,2-dicarbonyl reagent.

## Results

**Optimization and Synthesis of a 1,2-Dicarbonyl-forming Reagent.** We commenced our studies by investigating the transformation of readily available 2-hydroxy-1-phenylethanone to the corresponding α-ketothioester in the presence of $S_8$ and tetrabutylammonium bromide (TBAB) in cyclopentyl methyl ether (CPME) under an inert atmosphere. Unfortunately, desired α-ketothioester **2a** was not obtained when the reaction was run with only base or water (Table 1, entries 1, 2). **2a** could not be provided under the conditions of organic bases, regardless of whether water was added or not in the reaction (Table 1, entries 3–6). Encouragingly, dicarbonyl-forming reagent **2a** was isolated in 71% yield when both potassium carbonate and water were added (Table 1, entry 7). When potassium hydrogen carbonate ($KHCO_3$) was used instead of potassium carbonate ($K_2CO_3$), the yield increased to 86% (Table 1, entry 8). Decreasing the amount of water to 10 equivalents resulted in a lower yield (Table 1, entry 9). Increasing the equivalents of water did not improve the reaction outcome (Table 1, entry 10). TBAB was not necessary when DMF was used as the solvent, and the yield remained acceptable (Table 1, entry 11). When the reaction was carried out under air, the isolated yield of **2a** decreased to 61%, which means that oxygen affects this type of radical (Table 1, entry 12). However, the efficiency of the reaction was dramatically lower in the absence of TBAB as a phase-transfer reagent (Table 1, entry 13). The effects of different solvents implied the unique importance of cyclopentyl methyl ether (CPME) (see the Supporting Information).

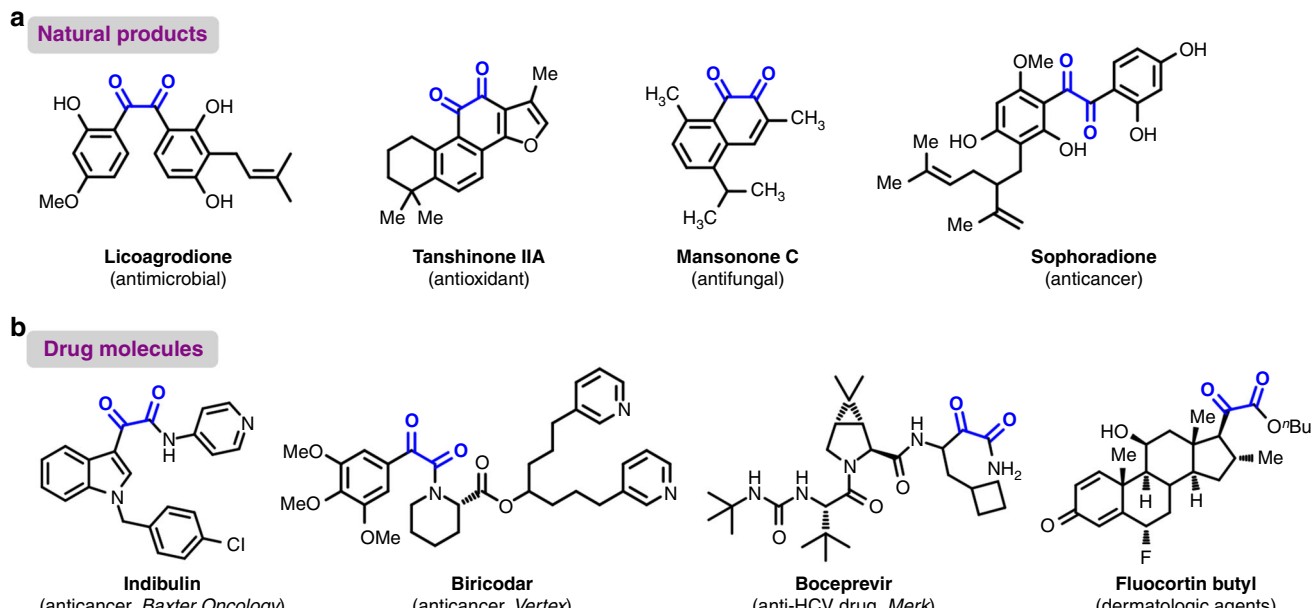

**Fig. 1** Significant dicarbonyl-containing molecules. **a** Dicarbonyl-containing natural products. **b** Dicarbonyl-containing drug molecules

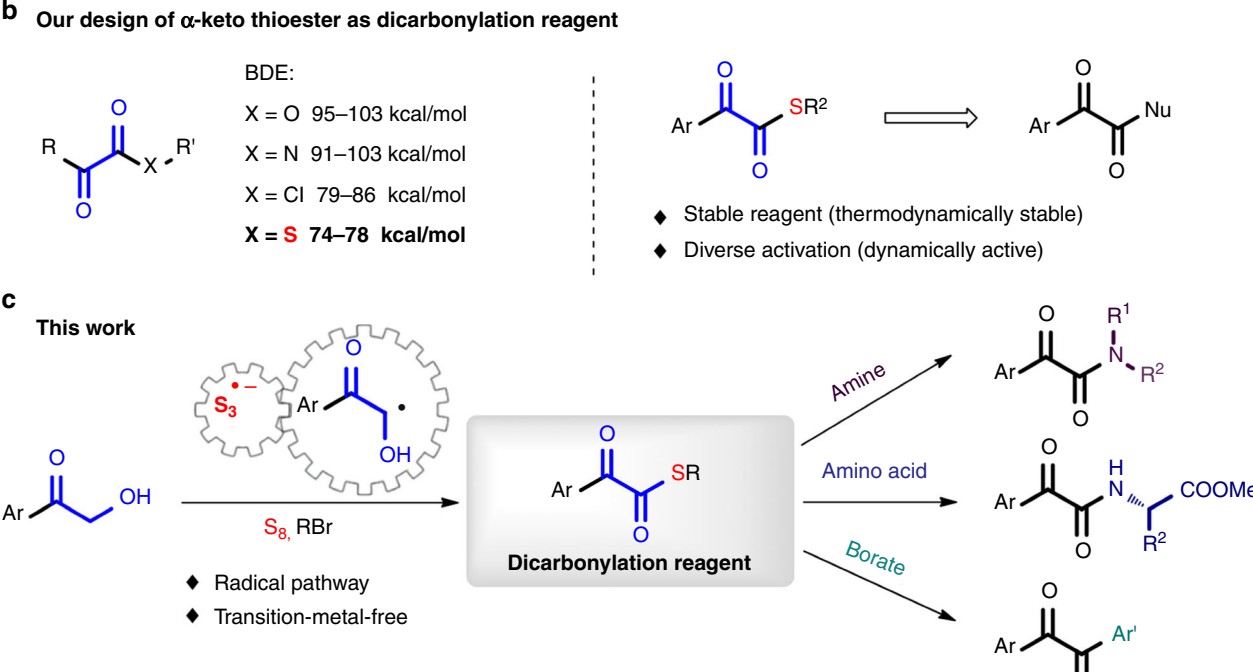

**Fig. 2** Strategies for the design of a 1,2-dicarbonyl-forming reagent and its applications. **a** The role of thioesters in life and biomimetic synthesis. **b** The strategy of α-ketothioester as dicarbonyl reagent. **c** Metal-free synthesis of an α-ketothioester reagent and its applications

Finally, $S_8$, $KHCO_3$ and TBAB in CPME with $H_2O$ under a $N_2$ atmosphere were chosen as the optimal conditions. The generality of the reaction was explored on a variety of α-hydroxy ketones, as illustrated in Table 2. α-Hydroxy ketones bearing electron-donating groups could be smoothly transformed into the desired products in moderate to good yields regardless of the position of their substituent (**2a**–**2e**). Notably, when the reaction was performed on a 10-mmol (1.36 g) scale of **1a**, product **2a** could still be obtained in a good yield (74%), which highlights the potential of α-ketothioesters as dicarbonyl transfer reagents. To our delight, α-hydroxy ketones bearing electron-withdrawing substituents on the aromatic ring gave moderate to good yields when DMF was used as the solvent (**2f**–**2k**), in which the generation rate and stability of S3·− radical and enolate radical anion can be improved in the high polar solvent. Sensitive groups,

such as ester and hydroxy groups, were also tolerated (**2k** and **2l**). Heteroaryl α-hydroxy ketones, such as 2-furyl and 2-benzofuryl derivatives, could also be converted to the corresponding α-ketothioesters in moderate yields (**2m**, **2n** and **2o**). With regard to alkyl bromides, both primary and secondary bromides were compatible with the reaction (**2p**–**2y**) as well terminal alkenes and alkynes (**2p** and **2r**). Benzyl bromides bearing different functional groups could be used to quench the reaction, affording the desired products in good yields (**2v**–**2y**).

The desired α-ketothioester **2a** could not be furnished when α-formyl ketones **1a′** was used as the substrate, which indicated that **1a′** is not the intermediate in this transformation (Fig. 3a). Radical-trapping experiments were conducted by investigating TEMPO under the standard conditions. The coupling product **2ab** was afforded in 35% yield, which provided strong evidence of

## Table 1 Optimization of the 1,2-dicarbonyl-forming reagent[a]

PhC(O)CH2OH (1a) → S8, Base, TBAB, H2O, CPME, 90 °C, then BnBr → PhC(O)C(O)SBn (2a)

| Entry | Base | $H_2O$ (equiv.) | Yields (%)[b] |
|---|---|---|---|
| 1 | — | 20 | NP |
| 2 | $K_2CO_3$ | – | Trace |
| 3[c] | $Et_3N$ | – | NP |
| 4[c] | DIPEA | – | NP |
| 5[c] | DBU | – | NP |
| 6[c] | $Et_3N$ | 20 | NP |
| 7 | $K_2CO_3$ | 20 | 71 |
| **8** | **$KHCO_3$** | **20** | **86** |
| 9 | $KHCO_3$ | 10 | 50 |
| 10 | $KHCO_3$ | 30 | 82 |
| 11[c,d] | $KHCO_3$ | 20 | 76 |
| 12[e] | $KHCO_3$ | 20 | 61 |
| 13[c] | $KHCO_3$ | 20 | 45 |

NP no product
[a]Conditions: **1a** (0.5 mmol), $KHCO_3$ (1 mmol), S8 (2.0 mmol), tetrabutylammonium bromide (TBAB) (0.1 mmol) and $H_2O$ (4 mmol) were stirred at 90 °C in cyclopentyl methyl ether (CPME) (4 mL) for 10 h under $N_2$, and then BnBr (0.75 mmol) was added. The system was heated for another 2 h under $N_2$
[b]Isolated yields
[c]Without TBAB
[d]DMF as solvent
[e]Under air

α-carbon radical being involved in this transformation (Fig. 3b). The coupling product **2ab** was not observed in the absence of elemental sulfur under the standard conditions, which indicate that sulfur is enssential for radical generation (Fig. 3c). A plausible reaction pathway that backed by these experimental evidences is described in Fig. 3d. The elemental sulfur interacted with the base to provide the trisulfur radical anion[31–33]. The α-hydroxy ketone can also initiate a persistent radical at the α position with the help of a base. Radical intermediate **I** was generated in the presence of both the trisulfur radical anion and a base. Subsequent coupling of the two different radicals in the reaction system afforded intermediate **II**. The dissociation of **II** could afford compound **III** and release HSS⁻, which combines with the alkyl halide to produce a mixture of polysulfide compounds (**BnSS$_n$Bn**). Intermediate **III** can tautomerize to sulfur anion **IV**, a more nucleophilic conformation, which can then undergo an $S_N2$ substitution to eventually form desired product **2**.

**Dicarbonylation with the Designed Reagents.** α-Ketoamides are an important fragment in drug discovery both in biological activities and synthetic transformations. A dicarbonyl fragment

## Table 2 Construction of a library of 1,2-dicarbonyl-forming reagents[a,b]

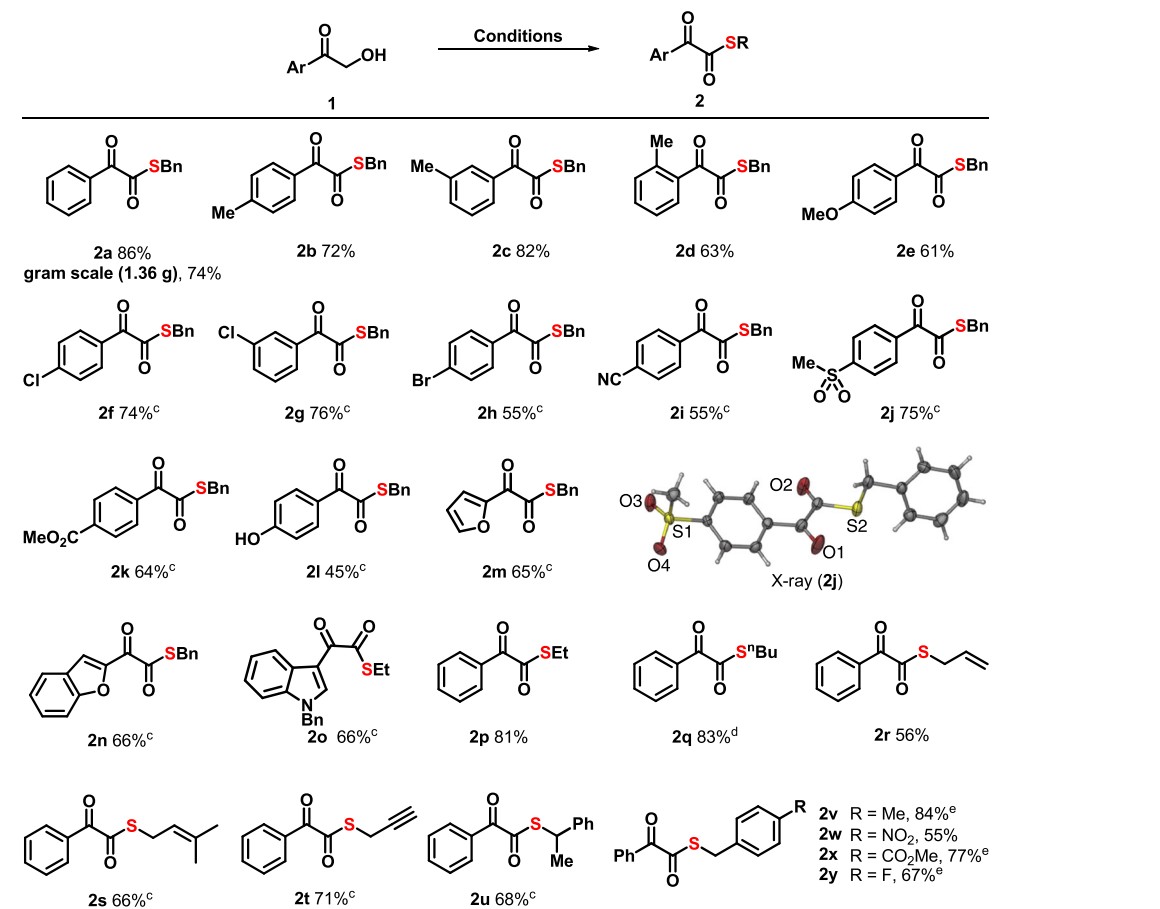

ArC(O)CH2OH (1) → Conditions → ArC(O)C(O)SR (2)

**2a** 86%
gram scale (1.36 g), 74%

**2b** 72%

**2c** 82%

**2d** 63%

**2e** 61%

**2f** 74%[c]

**2g** 76%[c]

**2h** 55%[c]

**2i** 55%[c]

**2j** 75%[c]

**2k** 64%[c]

**2l** 45%[c]

**2m** 65%[c]

X-ray (**2j**)

**2n** 66%[c]

**2o** 66%[c]

**2p** 81%

**2q** 83%[d]

**2r** 56%

**2s** 66%[c]

**2t** 71%[c]

**2u** 68%[c]

**2v** R = Me, 84%[e]
**2w** R = NO2, 55%
**2x** R = CO2Me, 77%[e]
**2y** R = F, 67%[e]

[a]Reaction conditions: **1** (0.5 mmol), $KHCO_3$ (1 mmol), S8 (2.0 mmol), TBAB (0.1 mmol) and $H_2O$ (4 mmol) were stirred at 90 °C in CPME (4 mL) for 10 h under $N_2$, and then RBr (0.75 mmol) was added. The system was heated for another 2 h under $N_2$
[b]10 mmol scale
[c]Reaction conditions: **1** (0.25 mmol), $KHCO_3$ (0.5 mmol), S8 (0.75 mmol), and TBAB (0.05 mmol) were stirred at 90 °C in DMF (2 mL) for 10 h under $N_2$, and then RBr (0.375 mmol) was added. The system was heated for another 2 h under $N_2$
[d]2.0 equiv. of $n$BuBr
[e]2.5 equiv. of RBr

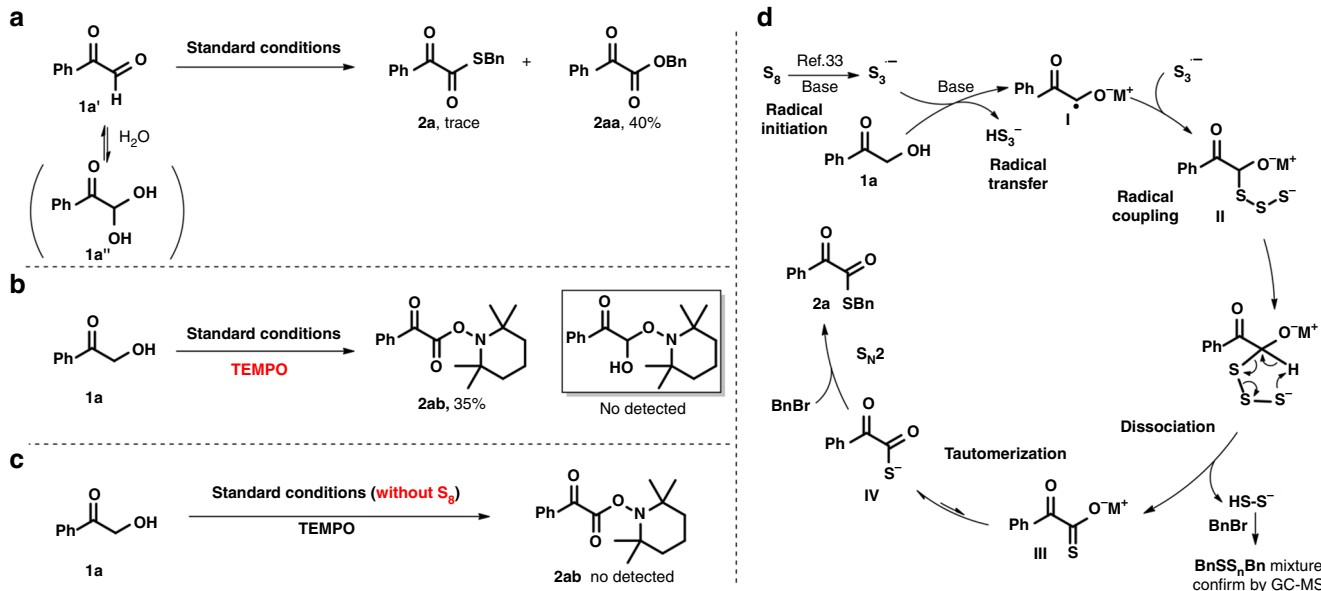

**Fig. 3** Mechanistic studies. **a** The control experiment of α-formyl ketones. **b** Radical-trapping experiments. **c** The control experiment in the absence of elemental sulphur. **d** The plausible reaction pathway

can be efficiently transferred to various primary or secondary α-ketoamides in good yields (including both aliphatic and aryl amines) under simple conditions (Table 3). 2-Oxo-2-phenylacetamide (**3a**), which possesses a free -NH₂ group, is an important intermediate in organic synthesis and showed various bioactivities. When ammonia gas was bubbled into a solution of **2a**, **3a** was obtained in 79% yield. Notably, L-phenyglycinol was successfully applied to the dicarbonylation protocol, and the hydroxyl group was tolerated (**3b**). Octadecylamine, a weak nucleophile, could be converted into the corresponding α-ketoamide in moderate yield (**3c**). In addition, the dicarbonylation of sterically hindered tert-butyl amine was successful, affording desired product **3f** in 57% yield. Anilines bearing a broad range of electron-donating and electron-withdrawing substituents at the para and ortho positions, the reactions proceeded smoothly (**3k-3n**). Secondary anilines were also compatible with this transformation, but in lower yield (**3o**). Due to the simple reaction conditions, i.e., no base, no metal and at room temperature, the dicarbonylation of a series of amino acid esters could afford the corresponding dicarbonyl compounds without erosion of the stereogenic information (>99% ee) in good yields (**3p-3x**). Notably, triglycine smoothly underwent the desired reaction to provide ethyl (2-oxo-phenylacetyl)glycylglycylglycinate (**3y**), which shows the great potential of this method for the late-stage modification of drug molecules.

Following the successful dicarbonylation of amines, the dicarbonylation of all-carbon aryl rings with borates for the construction of dicarbonyl-carbon bonds was studied. These reactions demonstrated the collective synthesis of dicarbonyl-containing derivatives via the current methodology (Table 4). The cross coupling of α-ketothioesters with a broad range of aryl borates was readily achieved to construct benzil derivatives. Aryl borates with substituents at different positions were all effective candidates, regardless of the electronic properties of the substituents and the presence of fused rings (**5a-5i**). Importantly, we demonstrated that 4-methoxyphenyl borates

can be successfully coupled with various 1,2-dicarbonyl reagents, including 2-furyl and 2-benzofuryl derivatives (**5j-5o**). It should be noted that the alkyl borate could be compatible in this transformation as well (**5p**).

The dicarbonylation of alcohols and water for the construction of dicarbonyl-oxygen bonds was further studied. Benzyl alcohol was successfully applied to the dicarbonylation protocol providing the desired product **2aa** in 73% yield. α-Ketothioester **2a** could be hydrolysed to 2-oxo-2-arylacetic acid **2ac** in the solution of sodium hydroxide. The common alcohols, such as ethanol and methanol, underwent smoothly affording the corresponding 2-oxo-2-arylacetate compounds in excellent yields (Fig. 4).

To further highlight the practical applicability of the 1,2-dicarbonyl-forming reagent, pharmaceutically relevant molecules and natural products were synthesized (Fig. 5). The anticancer drug indibulin could be prepared in a good yield from 1,2-dicarbonyl reagent **6a** through the dicarbonylation of the corresponding amine. The natural product polyandrocarpamide C and a 9,10-phenanthrenequinone derivative[34] were efficiently synthesized from 1,2-dicarbonyl reagents **6b** and **6c**, respectively, which represents a new synthetic route to these bioactive molecules.

## Discussion

In conclusion, a practical protocol for the straightforward construction of α-ketothioesters via the radical coupling of α-hydroxy ketones and elemental sulfur, in which S₈ was successfully introduced to a thioester, was disclosed. This method avoids the use of malodourous thiols. The application of practical α-ketothioester reagents for the dicarbonylation of amines and boroxine anhydride to afford α-ketoamide and benzil derivatives, respectively, was comprehensively achieved, and dicarbonyl motifs were successfully installed in these amino acid esters and peptides. These convenient reagents and procedures provide a potential method for the late-stage modification of drugs. Further studies on decarbonylations in drug discovery are ongoing in our laboratory.

**Table 3 Dicarbonylation of amines**[a]

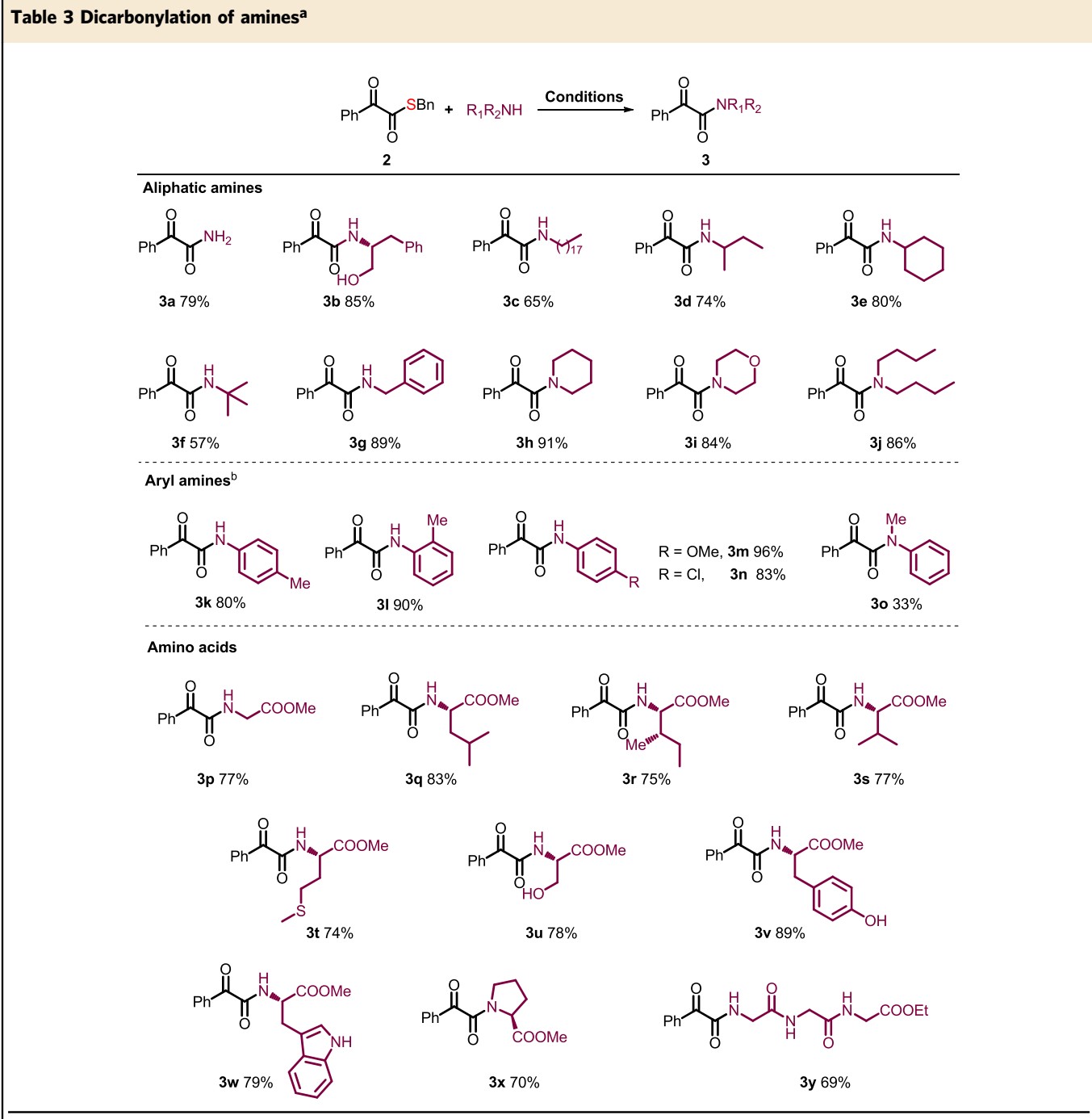

[a]Reaction conditions: **2** (0.2 mmol) and R[1]R[2]NH (0.2 mmol) in THF (2 mL) were stirred at room temperature under air. [b]DMAP (0.06 mmol) was added.

## Methods

**General procedure for syntheses of 1,2-dicarbonyl-forming reagents 2.** Under a $N_2$ atmosphere, α-hydroxy ketones **1** (0.5 mmol), $S_8$ (64.2 mg, 2 mmol), $KHCO_3$ (100 mg, 1 mmol), TBAB (32.3 mg, 0.1 mmol), $H_2O$ (180 mg, 4 mmol) and CPME (4 mL) were added to a Schlenk tube. After stirring for 10 h at 90 ℃ (detect by TLC), RBr (0.75 mmol, 1.5 equiv) was added to this mixture. The resulting mixture was allowed to stir for 2 h at 90 ℃. After completion of the reaction, water (5 mL) was added. The solution was extracted with ethyl acetate and organic layers were combined, dried over sodium sulfate before the organic phase was concentrated under vacuum. The residue was purified by column chromatography to give the corresponding product.

**General procedure for the dicarbonylation of amines.** α-Ketothioester **2** (0.2 mmol), amine (0.2 mmol) and THF (2 mL) were added to a reaction tube. After

stirring for 12 h at room temperature (detect by TLC), the solvent was removed and the residue was purified by column chromatography to give the corresponding product **3**.

**General procedure for the dicarbonylation of aryl borates.** Under a $N_2$ atmosphere, α-ketothioester **2** (0.1 mmol), aryl borate **4** (0.1 mmol), $Pd_2(dba)_3$ (0.0025 mmol, 2.5 mol%), 4,4'-dimethoxy-2,2'-bipyridine (0.01 mmol, 10 mol%), CuTc (0.1 mmol, 1 equiv), $K_2CO_3$ (0.15 mmol), anhydrous $Na_2SO_4$ (0.15 mmol) and DMF (1 mL) were added to a Schlenk tube. After stirring for 12 h at 45 ℃ (detect by TLC), the mixture was cooled to room temperature and water (5 mL) was added. Then the mixture was extracted with ethyl acetate and organic layers were combined, dried over sodium sulfate before the organic phase was concentrated under vacuum. The residue was purified by column chromatography to give the corresponding product.

## Table 4 Dicarbonylation of aryl borates[a]

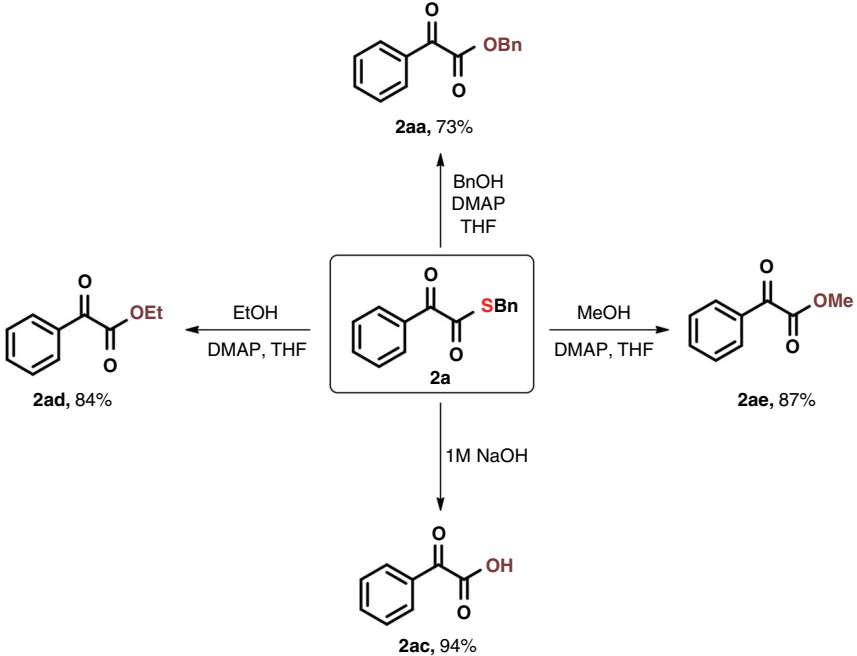

Reaction scheme and product array for entries 5a–5p.

**5a** 86%  **5b** 78%  **5c** 73%  **5d** 80%

**5e** 56%  **5f** 73%  **5g** 44%  **5h** 55%

**5i** 60%  **5j** 66%  **5k** 79%  **5l** 82%

**5m** 76%  **5n** 84%  **5o** 50%  **5p** 45%

[a]Reaction conditions: **2** (0.2 mmol), Ar(BO)$_3$ (0.2 mmol), Pd$_2$(dba)$_3$ (0.0025 mmol, 2.5 mol%), ligand (0.01 mmol, 10 mol%), CuTC (0.1 mmol, 1.0 equiv), K$_2$CO$_3$ (0.15 mmol, 1.5 equiv), Na$_2$SO$_4$ (0.15 mmol, 1.5 equiv) in DMF (1 mL).

**2aa,** 73%

**2ad,** 84%   EtOH / DMAP, THF   **2a**   MeOH / DMAP, THF   **2ae,** 87%

BnOH / DMAP / THF

1M NaOH

**2ac,** 94%

**Fig. 4** Further transformations. Dicarbonylation of alcohols and water

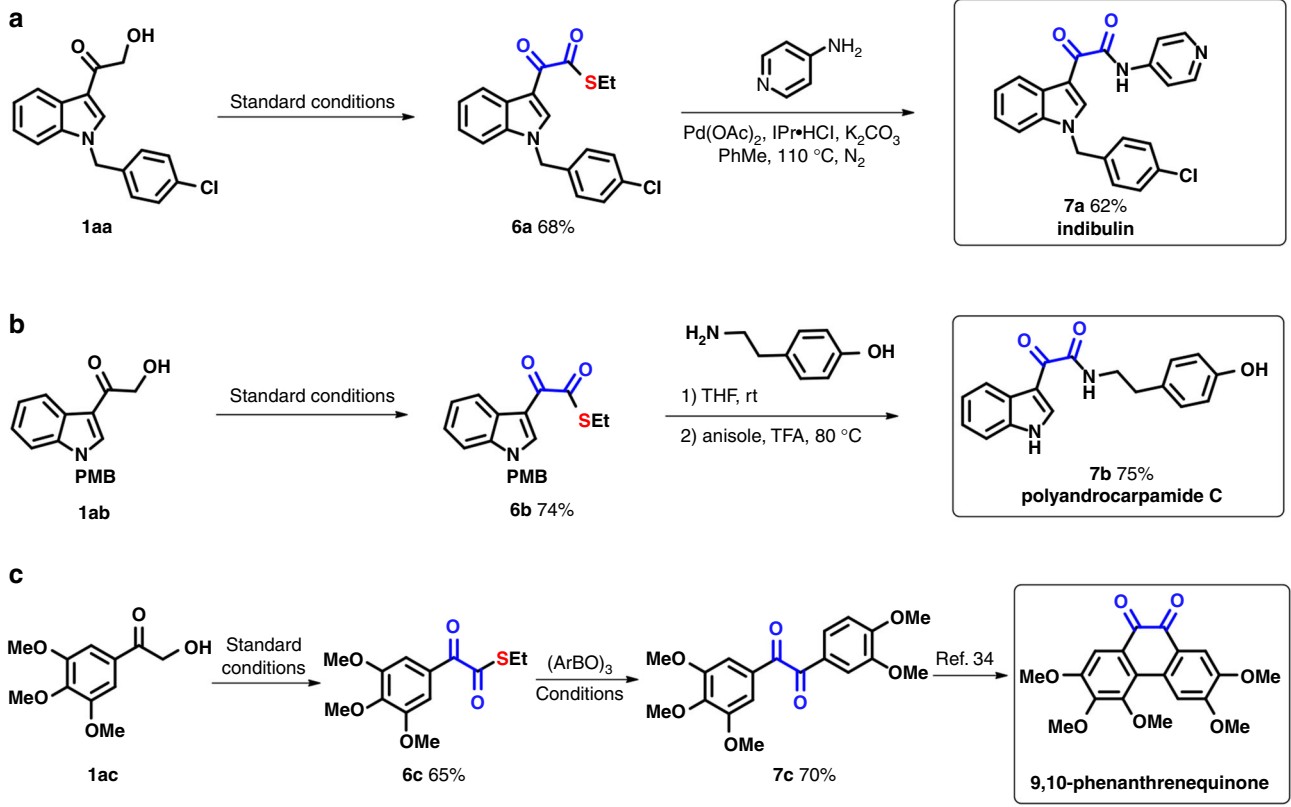

**Fig. 5** Synthesis of dicarbonyl-containing drugs and natural products. **a** The synthesis of indibulin. **b** The synthesis of polyandrocarpamide C. **c** The synthesis of a 9,10-phenanthrenequinone

## Data availability

The X-ray crystallographic coordinates for the structures reported in this study have been deposited at the Cambridge Crystallographic Data Centre (CCDC), under deposition number CCDC 1895971 (**2j**). These data can be obtained free of charge from the Cambridge Crystallographic Data Centre via www.ccdc.cam.ac.uk/data_request/cif. The authors declare that all other data supporting the findings of this study are available within the article and Supplementary Information files and are also available from the corresponding author on reasonable request.

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

## Acknowledgements

The authors are grateful for the financial support provided by The National Key Research and Development Program of China (2017YFD0200500), NSFC (21722202, 21672069; 21871089 for M.W.), S&TCSM of Shanghai (Grant 18JC1415600), Professor of Special Appointment (Eastern Scholar) at Shanghai Institutions of Higher Learning, and the National Program for Support of Top-notch Young Professionals.

## Author contributions

X.J. conceived the idea and supervised the whole project. M.W. and Z.D. designed and carried out the experiments. X.J. and M.W. discussed the results, contributed to the writing of the manuscript, and commented on the manuscript. All authors approved the final version of the manuscript for submission.

## Additional information

**Competing interests:** The authors declare no competing interests.

