## [Peer Review File · Nature Communications]

Reviewers' comments:

Reviewer #1 (Remarks to the Author):

This journal is aimed at publishing high impact, cutting edge new developments in Chemistry, and this paper is far too specialised for this. The basic idea has multiple exemplifications and methods, and this one, though novel, is only an iterative improvement and is an extension of expected chemistry. It would be more appropriate submitted to an organic journal.

Reviewer #2 (Remarks to the Author):

The present manuscript describes an efficient and practical protocol for the construction of α -ketothioesters via the radical coupling of α -hydroxy ketones and elemental sulfur, in which S₈ was successfully introduced to a thioester. Further mechanistic studies demonstrated that the trisulfur radical anion and the α -carbon radical of the α -hydroxy ketone are involved in this transformation. In addition, The dicarbonylation of a broad range of amines and amino acids and cross couplings with aryl borates to construct dicarbonyl-carbon bonds were realized by employing this α -ketothioester as a 1,2-dicarbonyl reagent. The work not only offers a practical and facile synthetic tool for useful compounds and the people interested in 1,2-dicarbonyl motif will benefit from this perspective, but also provide a potential method for the late-stage modification of drugs. It has enough novelty to be considered as a communication in Nature Communication, so I would like to recommend its publication.

However, some issues should be addressed before final acceptance:

1. The authors should make some comments on the representative experimental phenomena:

a. In Table 1, why the water is necessary? Why could the TBAB promote the reaction? Did the authors try to use organic base without water and TBAB?

b. Solvent has obvious influence on the reaction, and the authors should make some comments to explain the effect of solvents.

c. When α -hydroxy ketones bear electron-withdrawing substituents on the aromatic ring, the reaction have to changed conditions. This phenomenon indicated that the electronic effect of hydroxy ketones had significant influence on the reactivity, and the authors should illuminate the reason (stability or lifetime of radical).

d. For figure 2-B and control experiments on mechanistic study, the authors made some figure but no comments in the manuscripts.

2. The thioester will remove after the 1,2-dicarbonyl-forming reagents transform to other useful compounds. So, did the authors try to use other electrophilic reagents instead of only alkyl bromides to reduce the cost and toxicity or increase atom economy?

3. The authors demonstrated the practicability of α -ketothioester through many examples (table 3 and 4), but these examples were only simple extension of two kinds of reaction. More types of reactions will lead to more persuasion instead of rigid extension of single reaction.

Reviewer #3 (Remarks to the Author):

The article by Jiang and co-workers describes a transition-metal-free approach for the synthesis of an α -ketothioester reagent via the combination of an α -hydroxy ketone, elemental sulfur and a benzyl halide. This convenient process avoids the use of malodorous thiols and the corresponding products could be easily converted into 1,2-dicarbonyl compounds, which are commonly encountered in pharmaceuticals and biomolecules. The authors have also conducted some preliminary experiments to gain insights into the mechanism of the reaction. This is a good piece of work and I recommend its publication in Nature Communications after minor revisions as noted below.

1. The authors have performed the experiments well with aryl-substituted ketones. What happens with alkyl- or vinyl-substituted 2-hydroxy-ethanones? The 2-hydroxy-esters is also widely available, are they suitable for this transformation? The authors should comment on this in order to provide an idea about the scope of the reaction. If the reaction works with the substrates mentioned above, then a few more examples should be added to demonstrate the scope.

2. The authors have carried out some preliminary investigation about the mechanism in Fig. 3. These results should be described in the text (data is in the figure only). Also in the eq. 1 of Fig. 3, how does it work to generate the 2aa?

3. It takes two steps to generate the final 1,2-dicarbonyl compounds. Did the authors try the cascade or one-pot reaction? It would be potentially more useful if it can be done in one step to generate the corresponding amides and ketones.

4. In SI, some NMR spectrum is not pure, please purify them. For example, 2o, 3x, 7e.

5. In the NMR traces of Supplementary Information, a caption should be included on the NMR spectrum, noting the nucleus being measured, the solvent (formula preferred, e.g. CDCl₃ over Chloroform-d), and the field strength.

Responds to the reviewers' comments

Reviewer #1:

Q1: This journal is aimed at publishing high impact, cutting edge new developments in Chemistry, and this paper is far too specialised for this. The basic idea has multiple exemplifications and methods, and this one, though novel, is only an iterative improvement and is an extension of expected chemistry. It would be more appropriate submitted to an organic journal.

A1: This referee raised a reasonable point here that this journal is aimed at publishing high impact, cutting edge new developments in Chemistry. However, we do not agree with this referee that this paper is far too specialized, only an iterative improvement and an extension of expected chemistry. We would like to share our thoughts and rationale that justifies the publication of this work on *Nature Communications*.

1) These novel 1,2-dicarbonyl reagents can be easily constructed from safe, odorless and readily available inorganic elemental sulfur via a transition-metal-free approach. Mechanistic studies demonstrated that the trisulfur radical anion initiated the α -carbon radical of the α -hydroxy ketone in this transformation. This is also a new achievement for radical chemistry.

2) The dicarbonylation of a broad range of *N*-nucleophiles, *O*-nucleophiles, *C*-nucleophiles and even amino acids with sensitive chiral centers. Importantly, cross couplings with commercial available aryl borates to construct dicarbonyl-carbon bonds were first realized under mild conditions with this

stable and convenient α -ketothioester, which reveal the unique versatility and applicability of the 1,2-dicarbonyl reagents.

3) To further highlight the applicability of the 1,2-dicarbonyl reagents, pharmaceutically relevant molecules indibulin containing indolyl and pyridyl, natural products polyandrocarpamide C possessing free hydroxyl and a 9,10-phenanthrenequinone derivative with strong electron-rich aromatics were afforded with this reagent.

We believe that these 1,2-dicarbonyl reagents and broad-spectrum method will be of great interest for the field of organic chemistry, biochemistry and medicinal chemistry in general.

Reviewer #2: “recommend its publication”

Q1: The authors should make some comments on the representative experimental phenomena:

- a. In Table 1, why the water is necessary? Why could the TBAB promote the reaction? Did the authors try to use organic base without water and TBAB?
- b. Solvent has obvious influence on the reaction, and the authors should make some comments to explain the effect of solvents.
- c. When α -hydroxy ketones bear electron-withdrawing substituents on the aromatic ring, the reaction have to changed conditions. This phenomenon indicated that the electronic effect of α -hydroxy ketones had significant influence on the reactivity, and the authors should illuminate the reason (stability or lifetime of radical).
- d. For figure 2-B and control experiments on mechanistic study, the authors made some figure but no comments in the manuscripts.

A1: Thanks for the referee’s good suggestions. Comments have been added on the representative experimental phenomena. The changes of the manuscript are shown in “track of changes” form.

a. Water is necessary for the dissolution of inorganic base KHCO_3 . And TBAB is used as the phase transfer catalyst for the biphasic aqueous-organic solution (H_2O and CPME). No product was provided under the conditions of organic base without water and TBAB, due to the ready generation of enolate ion with help of water and inorganic base. (Please see the following Table).

Entry	Base	H_2O (equiv.)	Yields (%)
1 ^a	Et_3N	-	NP
2 ^a	DIPEA	-	NP
3 ^a	DBU	-	NP
4 ^a	Et_3N	20	NP
5	KHCO_3	20	86

^aWithout TBAB.

b. Different solvents were employed in different electrical property substrates for the current reaction. It is possible that solvent has a great influence on the generation rate and stability of $\text{S}_3^{\cdot-}$ radical anion and enolate radical anion.

c. α -Hydroxy ketones bear electron-withdrawing substituents on the aromatic ring shows weak reactivity. The reactions have to changed conditions (solvent) to improve the concentrate of $\text{S}_3^{\cdot-}$ radical anion. As this reviewer proposed, indeed, the generation rate and stability of $\text{S}_3^{\cdot-}$ radical can be improved when DMF or DMSO was used as the high polar solvent, which had been described by us and other groups [Ref. 31: *Nature*, **252**, 32-33 (1974); Ref. 32: *Chem. Soc. Rev.* **42**, 5996-6005 (2013); Ref. 33: *Org. Lett.* **18**, 5756-5759 (2016)].

d. Thanks for the reviewer's reminding. The descriptions about figure 2-B and control experiments on mechanistic study have been added in the manuscript. The changes of the manuscript are shown in "track of changes" form.

The description about figure 2-B was shown as “Due to the C-S bond possessing both weaker bond energy and relative stability at ambient conditions, we assumed that an α -ketothioester would be an excellent 1,2-dicarbonyl-forming reagent and be broadly applicable in chemistry (Figure 2-B)”.

The description about control experiments was shown as “The desired α -ketothioester **2a** could not be furnished when α -formyl ketones **1a'** was used as the substrate, which indicated that **1a'** is not the intermediate in this transformation (Figure 3-1). Radical-trapping experiments were conducted by investigating TEMPO under the standard conditions. The coupling product **2ab** was afforded in 35% yield, which provided strong evidence of α -carbon radical being involved in this transformation (Figure 3-2). The coupling product **2ab** was not observed in the absence of elemental sulfur under the standard conditions, which indicate that sulfur is essential for radical generation (Figure 3-3)”.

- Q2:** The thioester will remove after the 1,2-dicarbonyl-forming reagents transform to other useful compounds. So, did the authors try to use other electrophilic reagents instead of only alkyl bromides to reduce the cost and toxicity or increase atom economy?
- A2:** The simplest electrophilic reagent should be H^+ , which will generate the 1,2-dicarbonyl compound **A**. Unfortunately, compound **A** is not a suitable 1,2-dicarbonyl reagent, since $-SH$ group is more difficult to remove than $-SR$ group with strong coordination, strong smell and sensitive unstable property (exchange of O and S). Instead, the relative small electrophilic reagent Et^+ can afford the corresponding 1,2-dicarbonyl reagent **B** readily, which can not only provide a good atom economy, but also is a stable and easily converted 1,2-dicarbonyl transformation reagent.

A

B

Q3: The authors demonstrated the practicability of α -ketothioester through many examples (table 3 and 4), but these examples were only simple extension of two kinds of reaction. More types of reactions will lead to more persuasion instead of rigid extension of single reaction.

A3: Thanks for the referee's good suggestions. Besides N and C nucleophiles, dicarbonylation of different oxygen atoms were also realized under mild conditions employing this stable and convenient α -ketothioester as a 1,2-dicarbonyl reagent. The dicarbonylation of three types of alcohols and hydroxyl were achieved in excellent yields (see the following scheme).

Reviewer #3: recommended “Publish after minor revisions noted.”

Q1: The authors have performed the experiments well with aryl-substituted ketones. What happens with alkyl- or vinyl-substituted 2-hydroxy-ethanones? The 2-hydroxy-esters is also widely available, are they suitable for this

transformation? The authors should comment on this in order to provide an idea about the scope of the reaction. If the reaction works with the substrates mentioned above, then a few more examples should be added to demonstrate the scope.

A1: Alkyl- and vinyl-substituted 2-hydroxy-ethanones and 2-hydroxy-esters have been tried under many conditions but without promising result, probably due to the unstability of enolate radical ion.

Q2: The authors have carried out some preliminary investigation about the mechanism in Fig. 3. These results should be described in the text (data is in the figure only). Also in the eq. 1 of Fig. 3, how does it work to generate the 2aa?

A2: Thanks for the reviewer's reminding. The description about the experiments in Fig. 3 has been added in the manuscript. "The desired α -ketothioester **2a** could not be furnished when α -formyl ketones **1a'** was used as the substrate, which indicated that **1a'** is not the intermediate in this transformation (Figure 3-1). Radical-trapping experiments were conducted through investigating TEMPO under the standard conditions. The coupling product **2ab** was afforded in 35% yield, which provided strong evidence of α -carbon radical being involved in this transformation (Figure 3-2). The coupling product **2ab** was not observed in the absence of elemental sulfur under the standard conditions, which indicate that sulfur is essential for radical generation (Figure 3-3)."

For side product **2aa**, hydration of substrate **1a'** generates **1a''**, which affords the product **2aa** with coupling and oxidation under standard conditions. (reported before by our group: *Angew. Chem. Int. Ed.*, **2012**, *51*, 12570-12574.)

Q3: It takes two steps to generate the final 1,2-dicarbonyl compounds. Did the authors try the cascade or one-pot reaction? It would be potentially more useful if it can be done in one step to generate the corresponding amides and ketones.

A3: Thanks for the referee's good suggestions. Our goal is practical establishment of a stable and bench available 1,2-dicarbonyl reagent for divergent dicarbonylations for different systems. So, the combination of one 1,2-dicarbonyl reagent with different substrates (amine, amino acids, aryl borates and alcohols) could make this reagent more broad-spectrum, especially for drug discovery.

Q4: In SI, some NMR spectrum is not pure, please purify them. For example, **2o**, **3x**, **7e**.

A4: Thanks for the reviewer's reminding. New NMR spectrums compounds **2o** and **7c** (no **7e** in the current manuscript, this reviewer should mean **7c**) were provided in SI. The compound **3x** is pure actually, and it looks like with impurity since of rotamers, which have been noted in its NMR spectrums.

Q5: In the NMR traces of Supplementary Information, a caption should be included on the NMR spectrum, noting the nucleus being measured, the solvent (formula preferred, e.g. CDCl₃ over Chloroform-d), and the field strength.

A5: Thanks for the referee's good suggestions. The captions which include the nucleus, solvent and field strength have been added in all 77 compounds' ¹H NMR and ¹³C NMR spectrums (In Supplementary Information, Page 44-198).

REVIEWERS' COMMENTS:

Reviewer #2 (Remarks to the Author):

The authors made elaborate comments and supplements on the representative experimental phenomena. In addition, more types of reactions (Figure 4) have been proceeded to prove the practicability of α -ketothioester. So, I recommend the revised can be published as it is.

Reviewer #3 (Remarks to the Author):

The revision is very thorough and detailed. The authors answered all of my questions and also those of other reviewers. I hope to be able to follow up on this chemistry in the future. The manuscript should be accepted in this form.

Response to Reviews' Comments

Reviewer #2 (Remarks to the Author):

The authors made elaborate comments and supplements on the representative experimental phenomena. In addition, more types of reactions (Figure 4) have been proceeded to prove the practicability of α -ketothioester. So, I recommend the revised can be published as it is.

A: No revision is needed according to this referee.

Reviewer #3 (Remarks to the Author):

The revision is very thorough and detailed. The authors answered all of my questions and also those of other reviewers. I hope to be able to follow up on this chemistry in the future.

The manuscript should be accepted in this form.

A: No revision is needed according to this referee.